# Further Interpretation of the Volatile, Microbial Community and Edible Quality of Fresh Fermented Rice Noodles with Different Selected Strains

**DOI:** 10.3390/foods12050961

**Published:** 2023-02-24

**Authors:** Aixia Wang, Songfeng Xie, Zengrun Xia, Fengzhong Wang, Litao Tong

**Affiliations:** 1Key Laboratory of Agro-Products Processing, Ministry of Agriculture and Rural Affairs, Institute of Food Science and Technology, Chinese Academy of Agricultural Sciences, Beijing 100193, China; 2Product Research and Development Center, Chinese Academy of Se-Enriched Industry, Ankang 725000, China

**Keywords:** rice noodle, volatile components, microbial community, selected strain

## Abstract

Understanding bacteria and yeasts can reduce unpredictable changes in fresh fermented rice noodles (FFRN). The effects of selected strains (*Limosilactobacillus fermentum*, *Lactoplantibacillus plantarum*, *Lactococcus lactis* and *Saccharomyces cerevisiae*) on the edible quality, microbial community, and volatile component of FFRN were studied. The results indicated that the fermentation time could be shortened to 12 h when *Limosilactobacillus fermentum*, *Lactoplantibacillus plantarum*, and *Lactococcus lactis* were added, whereas it still required approximately 42 h after adding *Saccharomyces cerevisiae*. Only a steady bacterial composition was provided by adding *Limosilactobacillus fermentum*, *Lactoplantibacillus plantarum*, and *Lactococcus lactis*, and only a steady fungal composition was provided by adding *Saccharomyces cerevisiae*. Therefore, these microbial results indicated that the selected single strains cannot improve the safety of FFRN. However, the cooking loss was decreased from 3.11 ± 0.11 to 2.66 ± 0.13 and the hardness of FFRN was increased from 1186 ± 178 to 1980 ± 207 when it was fermented with single strains. Finally, a total of 42 volatile components were determined by Gas chromatography-ion Mobility Spectrometry and 8 aldehydes, 2 ketones, and 1 alcohol were added during the entire fermentation process. The main volatile components were different during fermentation depending on the added strain, and there was the greatest variety of volatiles in the group with added *Saccharomyces cerevisiae*.

## 1. Introduction

Fresh fermented rice noodles (FFRN) are usually fermented spontaneously in most factories since ancient times, which has been considered a conventional and essential process to enhance sensory properties and texture profiles [1]. Nevertheless, the metabolic activity of complex and diverse microorganisms in the environment and raw materials during natural fermentation always results in inconsistencies of texture and sensory properties among batches and even food safety risks [2,3]. In response to these problems, the usage of starter cultures has substituted natural fermentations to offer standardized properties of FFRN [4,5,6]. It has been well established that these starter cultures consist of different species and that they are positively conducive to the fermentation process by improving the safety and by optimizing organoleptic characteristics. In general, different microbes play their corresponding roles during each fermentation stage, which include initially multiplying via aerobic respiration, followed by various complex biochemical processes via anaerobic respiration to form a low pH environment, pleasant texture profiles, and desired flavor compounds [7,8].

Recently, the presence of pathogenic bacteria in each of the 10 FFRNs with spontaneous fermentation from our laboratory in southern China had been addressed in our previous study [9]. In addition to the food safety of the FFRN, flavor is also a crucial indicator for consumer acceptance, so it is necessary to characterize the microbiota succession and flavor changes in rice noodles during fermentation [10,11]. Headspace-gas Chromatography-ion Mobility Spectrometry (HS-GC-IMS), a burgeoning and robust technique, possesses the advantages of both the well separation efficiency of GC and the high sensitivity of IMS. Meanwhile, it is equipped with an automatic headspace injector that could qualitatively detect the volatile compounds from a liquid or solid sample without requiring any pretreatment, which makes it convenient [12]. Therefore, it has been widely utilized in many areas, such as analyzing the flavor of tomato pastes [13], the freshness of eggs [14], and dynamic changes in green tea [15].

However, little information is available about the effect of individual microbes that comprise starter cultures on the flavor and safety of FFRN. In particular, the composition of the strain species for some commercial starter cultures is not clear [5]. Therefore, exploring the association of each microbiota with property promotion and safety assurance is essential to understand, control, and improve fermentation efficiency to ensure the quality and safety of FFRN.

Therefore, we explore the crucial role of each microbiota for safety, flavor, and edible quality of FFRN by using our previous starter culture as a model system in this study. Firstly, four strains (*Limosilactobacillus fermentum*, *Lactoplantibacillus plantarum*, *Lactococcus lactis* and *Saccharomyces cerevisiae*) were applied in order to ferment the rice noodle. The cooking quality, texture profile, microbial community, and volatile component of the FFRNs were then compared. This study aims to ascertain the influence of each strain on these properties and the relationship between them. We hope that the results of the present study will provide a theoretical basis for ensuring the safety and quality of FFRN during fermentation and for enhancing the industrial productivity of FFRN.

## 2. Materials and Methods

### 2.1. Materials

Zhenzhu 3 Indica rice was collected from Nanling market, Anhui province, China. *Limosilactobacillus fermentum* 22538, *Lactococcus lactis* 20712, and *Saccharomyces cerevisiae* 1340 were collected from the China Center of Industrial Culture Collection. *Lactoplantibacillus plantarum* YI-Y2013 (CCTCC NO: M2017533) was collected from the China Center for Type Culture Collection.

### 2.2. Activation of Strains

*Limosilactobacillus fermentum*, *Lactoplantibacillus plantarum*, and *Lactococcus lactis* were cultivated in MRS Broth medium (AOBOX, Beijing AOBOX Biotechnology Co., Ltd., Beijing, China) at 37 °C and *Saccharomyces cerevisiae* was cultivated in YM Broth medium (AOBOX, Beijing AOBOX Biotechnology Co., Ltd.) at 28 °C until the strain concentration reached 10^8^ cfu/mL. The resultant cultures were centrifuged at 10,000× *g* at 4 °C for 15 min and the sediment was dissolved in 0.85% sterilized normal saline. The process was repeated to thoroughly remove residual broth medium. Finally, the strains were collected and suspended in normal saline at a concentration of 10^8^ cfu/mL [15].

### 2.3. Preparation of FFRNs

0.5% pure strain suspensions were added respectively into the rice slurries that contained rice flour and sterile water in a ratio of 11:9 (g:g) in conical flasks and the mixtures fermented at 28 °C [6,16]. Meanwhile, another sample of rice slurry with the same mass ratio of rice flour and sterile water was fermented in a natural environment. This rice slurry was used as the control group and was fermented under the same conditions as the rice slurries with pure strains. All rice slurries were taken out at a predetermined time (0 h, 6 h, 12 h) and stored at 4 °C for experimental analysis.

The processing and storage of fresh fermented rice noodle is according to our previous research with some modifications: the five groups of fermented rice slurries were poured into plates and steamed at 100 °C for 10 min. The steamed rice sheets were kneaded into rice dough and extruded into rice noodles using a rice noodle machine (MOB50 × 150, Xingtai, China). These fresh fermented rice noodles were cooked for 60 s in boiling water and cooled to room temperature using deionized water. The water on the surface of the rice noodles was drained in the air, and the noodles were stored in stainless steel plates at 4 °C for further analysis. 

### 2.4. The pH Value and Proximate Composition Analysis

In order to inspect the pH values of the rice slurries during fermentation, a pH meter (pH 400, Shanghai, China) was used. The total starch content in the fresh fermented rice noodle samples was determined by a Total Starch Assay Kit (K-TSTA-50A/K-TSTA-100A 06/17, Megazyme International Ltd., Wicklow, Ireland) from the American Association for Clinical Chemistry (AACC) approved method 76-13.01. The protein content was determined with a Kieltec analyzer (Foss Tecator AB, Höganäs, Sweden), using the Kjeldahl method with a conversion factor of 5.95 (AOAC, 984.13). The lipid content was determined using the analytical method provided by the Association of Official Agricultural Chemistry (AOAC, 945.16) [9].

### 2.5. Cooking Qualities and Textural Properties Analysis

Cooking properties of FFRNs were detected according to a method with some modifications [17]. Briefly, 50 g of rice noodles were boiled in 500 mL of deionized water for 2 min and then weighed after draining for 5 min. The cooking water was collected to measure its turbidity at 675 nm and then dried in an oven at 105 °C to a constant weight. Water absorption and cooking loss were calculated by the following formulae.
Water absorption = (Weight of cooked rice noodles − Weight of uncooked rice noodles)/Weight of uncooked rice noodles.
Cooking loss = Dry weight of the cooking water/Weight of uncooked rice noodles.

The method of texture analysis was provided by our previous study with some modifications [9]. Three fresh fermented rice noodle samples were prepared and tested at equal intervals on the test bench by a P/36R probe (36 mm diameter cylindrical probe). Five replicates of each treatment were measured, and the values were averaged.

### 2.6. Volatile Components Analysis

The volatile components of all samples were determined by GC-IMS (Flavourspec^®^, G.A.S, Dortmund, Germany), according to the method of [15]. Two-gram FFRN samples were directly transferred into a 20 mL flask and incubated at 80 °C for 15 min. Next, a syringe at 85 °C automatically injected 500 μL of headspace and put the sample into the MXT-WAX capillary column (30 m × 0.25 mm × 0.25 μm; Agilent Technologies, Santa Clara, CA, USA) under 60 °C isothermal conditions. The program used was: 2 mL/min for 2 min, 10 mL/min for 8 min, and 100 mL/min for 20 min until flow stopped. The volatile substances were driven to the ionization chamber after gas chromatographic separation and were ionized in a positive ion mode. The drift tube was 98.0 mm long, with a drift gas flow rate of 150 mL/min. The temperature of the column and the drift tube were kept at 60 °C and 45 °C, respectively. The standard solution of normal ketones (C4–C9) was tested under the same gas phase conditions as the sample, and the calibration curves of the retention index (RI) and retention time (Rt) were established. The corresponding retention index of unknown substances could be calculated through the retention time. The analytical spectrum was viewed using the Laboratory Analytical Viewer. The visual and quantitative comparisons of volatile differences between different samples were performed by using the Gallery plot plug-in. Two-dimensional top view and three-dimensional fingerprint maps were constructed by using the Reporter plug-in. The contribution of each volatile component to the flavor of FFRN was estimated by calculating the relative odor activity value (ROAVi) [18]. The compound that contributed the most to the flavor of all samples was defined as ROAVm with a value of 100. Other volatile compounds were calculated as follows:ROAVi = 100 × Ci/Cm × Tm/Ti 
where Ci was the relative concentration (the ratio of the peak intensity of each compound compared to the total peak intensity of all compounds), and Ti expressed the odor thresholds of the target volatile substances. Cm and Tm represented volatile components with maximum odor activity value.

### 2.7. Microbial Composition Analysis

The detailed DNA extraction, PCR amplification and further methods are referenced in our previous methods [6,19]. The sequences were clustered into the same operational taxonomic units (OTUs) at a similarity level above 97% (USEARCH, version 10.0). The OTUs were filtered with a threshold of 0.005% for all sequence numbers [20].

### 2.8. Statistical Analysis

Every experiment was performed three times before the final experiment and each experiment was operated in triplicate. The data were expressed as the mean ± standard deviation (SD) and analyzed by SPSS26.0 using Tukey Kramer’s multiple comparison post hoc tests, with a significance level of 95% (*p* < 0.05). The significance analysis between groups was based on one-way analysis of variance (ANOVA) by using SPSS statistics 26.0 software (SPSS Inc., Chicago, IL, USA). Qualitative analysis of volatile components was performed based on the built-in database of GC × IMS Library Search. Alpha diversity (Coverage, Chao richness and Shannon diversity indices) was evaluated using the Mothur v.1.30.1 program, and Beta diversity (Hierarchical clustering, Principal Component Analysis) was accompanied by the ANOSIM statistical analysis based on Bray-Curtis distances and performed using QIIME v. 2-2022.2 software.

## 3. Results and Discussion

### 3.1. The pH Value

The change in pH (from 0 h to 48 h) in the fermented rice paste is depicted in Figure 1A. It first decreased before maintaining a steady level. Compared to *S. cerevisiae* and NF (fermented naturally), the other three groups presented a lower pH value. In detail, the pH value of the *L. fermentum*, *L. plantarum* and *L. lactis* groups reduced faster (12 h) to a stable value (pH = 4.1), The pH of the S. *cerevisiae* group decreased moderately until it remained constant at 5.0, while natural fermentation brought the pH value of the rice slurries down to 5.0 in 42 h. These results indicate that the *Lactobacillus* family, including *L. fermentum*, *L. plantarum* and *L. lactis*, could produce lactic acid to reduce the pH value by breathing in anaerobic environments.

### 3.2. The Content of the Total Starch, Protein and Lipid

The total starch, protein and lipid contents in FFRNs with different treatments are exhibited in Figure 1B–D. The total starch content in these five groups had no significant difference, which coincided with the study of [16]. Compared with RF, protein content in the *L. fermentum* group decreased the most, followed by the *S. cerevisiae*, *L. lactis*, *L. plantarum*, and NF groups. The lipid content in the *S. cerevisiae* group reduced significantly, followed by the *L. fermentum*, *L. lactis*, *L. plantarum*, and the NF groups, possibly because the microorganisms decomposed the lipid into fatty acids. These results indicated that *Lactococcus lactis* shows the best ability to transform starch, *Lactobacillus fermentum* shows the best ability to transform protein, and *Saccharomyces cerevisiae* shows the best ability to transform lipid. In total, the fermentation process can promote the conversion of starch, protein and lipid into other compounds, and these four strains exhibited different abilities to convert them, which would be the main reason for flavor differences.

### 3.3. The Cooking Qualities and Textural Profiles

The cooking and texture properties are important indicators to evaluate the edible quality and consumer acceptance of rice noodles [2]. As shown in Table 1, the cooking quality and texture profiles of the five groups are presented. The water absorption rate of the *S. cerevisiae* group was shown to be the highest, followed by the NF group, the *L. fermentum* group, the *L. plantarum* group, and finally the *L. lactis* group. This was possibly due to the internal structure of the lowest protein content (more than 90% hydrophobic protein accounts for rice protein) of the *S. cerevisiae* group noodles being able to hold more water [21]. These results indicated that the water absorption rate is related to the interaction between water and other components (total starch and protein) in the FFRNs [22]. Cooking loss and cooking water turbidity have always served as an indicator of the structural network of FFRNs. A lower cooking loss and cooking water turbidity indicates a better quality of FFRNs [23]. The *L. plantarum*, *L. fermentum*, *L. lactis*, and *S. cerevisiae* groups showed lower cooking qualities than the NF group and the sample of *L. lactis* exhibited the lowest one. These results indicated that adding a single strain could improve the cooking quality of rice noodles, which is in consensus with our previous study [6].

The hardness of FFRNs was clearly increased by adding different strains, and this is in agreement with the previous conclusion [6]. Many studies have reported that lipids could encompass amylose, resulting in the hardening of the rice noodle texture as the lipid content was reduced and the amylose helix exposed. Therefore, amylose was an important factor in the formation of the starch gel network, and it is a pity that the amylose content was not measured in this study. Moreover, *Saccharomyces cerevisiae* could transform monosaccharides to CO_2_, alcohol and flavor compounds in the fermentation process, which formed some holes in FFRNs. That is why the hardness of the *S. cerevisiae* group was lower than that of the *L. fermentum*, *L. plantarum* and *L. lactis* groups [7]. With the addition of a single strain, the resilience and chewiness also showed an increase, similar to the increase in hardness, indicating that rice noodles fermented with added strains had a better texture that consumers desire [5]. These results demonstrated that the fermentation of *Lactobacillus*, especially *L. fermentum*, *L. plantarum* and *L. lactis* was an important factor that had an impact on the texture profile.

### 3.4. Dynamic Analysis of Microbial Community

#### 3.4.1. Dynamic Analysis of Bacterial Community

The sequencing data were analyzed at two levels to study the bacterial community of FFRNs in the fermentation process (Figure 2A,B). The results demonstrated that a total of three bacteria were annotated at phyla level, including *Firmicutes*, *Cyanobacteria* and *Proteobacteria*. The *Firmicutes’* relative abundance of *L. plantarum*, *L. fermentum* and *L. lactis* groups was increased significantly (ranging from 41.65% to 94.65%), and it increased from 9.82% to 77.56% in the *S. cerevisiae* group during the entire fermentation process. While the highest relative abundance of *Firmicutes* in the NF group was shown to be at the 6 h stage (79.57%), and there were no significant differences between the 12 h stage (44.43%) and the 0 stage (41.64%) in the *L. plantarum* group. The relative abundance of *Cyanobacteria* in all samples reduced significantly during the entire fermentation process (ranging from 74.73% to 3.19%). The relative abundance of *Proteobacteria* in the *L. plantarum*, *L. fermentum* and *L. lactis* groups decreased (ranging from 18.52% to 0.30%) during fermentation while that of the NF group increased from 15.09% to 52.27%. The composition and variation trend of the bacteria at the phyla level of FFRNs with these selected strains was similar to that of FFRNs with starter culture [6].

Nine bacterial genera were annotated by bacterial sequencing data depicted in Figure 2B. The bacterial community composition of FFRN showed significant diversity. The proportion of *Lactobacillus* (ranging from 40.73% to 94.12%) instead of *Norank-f-norank-Chloroplast* (ranging from 53.44% to 5.08%) became dominant in the *L. plantarum*, *L. fermentum* and *L. lactis* groups as the fermentation time increased. Similarly, *Norank-f-norank-Chloroplast* (72.50%) was replaced by *Lactobacillus* (62.69%) and *Exiguobacterium* (13.12%) at the 12 h stage in the *S. cerevisiae* group, while *Norank-f-norank-Chloroplast* (74.73%) was replaced by *Exiguobacterium* (75.33%) in the NF group. Finally, *Lactobacillus* increased remarkably (ranging from 6.70% to 94.12%) while *Norank-f-norank-Chloroplast* decreased noticeably (ranging from 72.50% to 3.20%) in the *L. plantarum*, *L. fermentum*, *L. lactis*, and *S. cerevisiae* groups throught the entire fermentation process. Lactobacillus was dominant in the L. plantarum, L. fermentum, L. lactis, and S. cerevisiae groups at the final stage, and *Exiguobacterium* (42.20%), *Unclassified-f-Enterobacteriaceae* (33.37%), and *Cronobacteria* (19.23%) formed the bacterial composition skeleton of the NF group. The bacterial compositions of the surrounding environment and the material were various and multiple, which strongly influenced the bacterial composition of the NF FFRNs. Among the bacteria of the NF group, *Exiguobacterium*, *Cronobacter* and *unclassified_f_Enterobacteriaceae* were generally regarded as the main spoilage bacteria, and resulting infections from these would lead to intestinal or urinary tract problems [24,25]. Therefore, the quality of the FFRNs fermented spontaneously was not guaranteed [6,9]. The results showed that *Norank-f-norank-Chloroplast* was predominant in all samples at the 0 h stage but were quickly replaced by *Lactobacillus* in the *L. plantarum*, *L. fermentum*, and *L. lactis* groups during fermentation. This result demonstrated that the low pH environment that was supplied by *Limosilactobacillus fermentum*, *Lactoplantibacillus plantarum*, and *Lactococcus lactis* also could enhance the safety of FFRN by inhibiting the growth of *Norank-f-norank-o-Chloroplast* and other spoilage bacteria during fermentation.

#### 3.4.2. Dynamic Analysis of Fungal Community

The sequencing data were analyzed at two levels to study the fungal community of FFRNs at 0-h, 6-h and 12-h stages (Figure 3A,B). Figure 3A indicated that three fungi were determined at the phyla level, including *Ascomycota*, *Baidiomycota*, and *Unclassified-Fungi*. The *Ascomycota* was predominant at the early stage and increased moderately (ranging from 85.57% to 99.99%) in all samples from 0 h to 12 h. In total, there was no significant difference in fungal composition at the phyla level between the NF group and the groups with added strains.

As shown in Figure 3B, 28 genera were annotated by fungal sequencing data. The fungal composition of the *L. plantarum*, *L. fermentum*, *L. lactis*, and NF groups was varied and complex, and there was no change during the entirety of fermentation, centrally including *Curvularia*, *Aspergillus*, *Fusarium*, *Gibberella*, and *Nigrospora*. The fungal composition of the *S. cerevisiae* group was diverse and *Saccharomyces* (73.99%) dominated at the 0 h stage, whereas *Saccharomyces* increased significantly (99.47%) and restrained the growth of other fungi. The fungal diversity in the surrounding environment was diverse and uncontrollable, which also strongly influenced the fungal composition of the NF FFRNs. Among the fungi of the *L. plantarum*, *L. fermentum*, *L. lactis* and NF groups, *Curvularia* were saprophytic dematiaceous fungi generally isolated from environmental sources and they were answerable to allergic fungal rhinosinusitis [26], *Aspergillus* infections occurred in the form of a variety of respiratory infections [27], which led *Curvularia* and *Aspergillus* to be identified as human pathogens. Therefore, the aspect of fungi also demonstrated that the safety of the spontaneously fermented FFRNs was not guaranteed.

In total, the bacterial diversity of the *L. plantarum*, *L. fermentum* and *L. lactis* groups declined while fungal diversity was not improved. Similarly, the fungal diversity of the *S. cerevisiae* group declined while the bacterial diversity of the *S. cerevisiae* group was also not improved. These results meant that the separate addition of fungi (*Saccharomyces cerevisiae*) and bacteria (*Lactobacillus fermentum*, *Lactobacillus plantarum* and *Lactococcus lactis*) could improve the safety of FFRN by inhibiting the corresponding bacteria and yeast [28].

#### 3.4.3. Differences and Similarities in Fermentation

The Venn chart could summarize the similarities and differences of five groups. In Figure 2C,D, the bacterial compositions at the phylum and genus level of FFRNs fermented with four selected strains are depicted. A total of five bacteria at phylum level and 44 bacteria at genera level were detected during these three fermentation stages. Three bacteria at phylum level and eight bacteria at genera level were detected at each stage among these bacteria, which indicated that these common bacteria played an important role throughout the entire fermentation, while thirteen, four, four, six, two, eleven and one specific bacteria at genus level were detected in the *L. plantarum*-0, *L. lactis*-0, NF-0, *L. plantarum*-6, *L. Lactis*-6, *L. fermentum*-6 and NF-12 groups, respectively. These specific bacteria meanly exited in the first and second stage of the fermentation in the *L. plantarum*, *L. lactis* and *L. fermentum* groups, especially at the last stage of fermentation in the NF group. The results indicated that the bacterial composition of the FFRNs in the NF group was complex and uncertain, which would affect the safety of the FFRN. These results agreed with that of the dynamic analysis of the bacterial community (Section 3.4.1).

The fungal compositions at the phylum and genus level of FFRNs with different strains are shown in Figure 3C,D. A total of three fungi at phylum level and 21 fungi at genus level were tested during the entire fermentation process. Two fungi at phylum level and 21 fungi at genus level were tested at each stage among these fungi, which indicated that these common fungi played an important role in the whole fermentation process. While there were 11, 7, 8, 3 and 11 fungi tested in the *L. plantarum*-0, *L. fermentum*-0, *L. lactis*-0, *S. cerevisiae*-0 and NF-0 groups, respectively; 6, 3, 5, 2 and 13 fungi were detected in *L. plantarum*-6, *L. fermentum*-6, *L. lactis*-6, *S. cerevisiae*-6 and NF-6, respectively; there were 3, 15, 16, 2 and 7 fungi detected in *L. plantarum*-12, *L. fermentum*-12, *L. lactis*-12, *S. cerevisiae*-12 and NF-12, respectively. These specific fungi meanly exited in the first and second stages of fermentation in the *L. plantarum*, *L. lactis* and *L. fermentum* groups, especially the entire fermentation process in the NF group. The results indicated that the fungal composition of the FFRNs in the NF group was complex and uncertain, which would affect the safety of the FFRN. These results agreed with that of the dynamic analysis of the fungal community (Section 3.4.2).

Linear discriminant analysis effect size (LEfSe) with Linear discriminant analysis (LAD)score > 4 was used to analyze the fungi and bacteria that resulted in differences and similarities among the five groups. The LDA diagram displays the microbiota as classified from the phyla level to the genus level. As depicted in Figure 2E, the NF-0 group and the NF-12 group showed the most varied bacteria, mainly consisting of two phyla, three classes, four orders, five families, and six genera. Meanwhile, the NF-12 group also possessed the most varied fungi, mainly consisting of five classes, ten orders, fifteen families, and twenty-one genera (Figure 3E). The *L. lactis*-12 group possessed some unique bacteria, mainly consisting of one phylum, one class, four orders, four families, and four genera, while the *L. plantarum* and *L. fermentum* groups did not show up. The presence of these distinctive bacteria is mainly due to the addition of *Lactococcus lactis*, which meant that different *Lactobacillus* had different effects on the final bacterial composition of FFRN.

#### 3.4.4. Correlation of Occurrence Patterns between the Microbiological Communities

In order to elucidate the correlation of co-occurrence and co-exclusion among bacteria and among fungi, the correlation network at a genus level was investigated with Spearman. The red color represents a positive relation and the green represents a negative relation, the thickness of the connecting line between microbes represents the degree of correlation. As shown in Figure 4A, *Lactobacillus* co-excluded with *Acinetobacter*, *Cronobacter*, *Enhydrobacter*, *unclassified_f__Enterobacteriaceae* and *Exiguobacterium*. Figure 4B showed that *Saccharomyces* co-excluded with *Penicillium*, *Nigrospora* and *Aspergillus*. These results indicated that the addition of *Lactobacillus* and *Saccharomyces* could restrain some spoilage microbes, which is consistent with the dynamic succession of the microbiological community.

### 3.5. Changes of Volatile Components

The changes of volatile components of FFRN with four selected strains at 0-h, 6-h and 12-h stages were detected by HS-GC-IMS, which are presented in the fingerprints where the row represents the compounds and the column represents the sample name. The red circled regions (A to K) contain the defined differential volatile compounds among groups. As shown in Figure 5, the type of volatile compound was increased after formation, and the volatile compounds of FFRNs with different strains at these three stages were significantly different except at the 0 h stage. At the first stage (circled region A), the peak volume of 2-methylbutanal, 2-nonanone, 2-heptanone, E-2-nonenal, n-nonanal, hexyl acetate, E-2-heptenal, 1-pentanol, and pentanal was relatively higher in five groups, where these volatile compounds mainly come from the rice flour. As the fermentation time goes by, the peak volume of 1-heptanol, 2-pentanone, 3-methyl butanol, 1-hexanol, 2-nonenal, and 2-heptanone was relatively higher in the *L. plantarum*-6 group (circled region B); the peak volume of 1-hexanol, E-2-nonenal, 1-heptanol, 3-methyl butanal, 2-methyl butanal, 2-heptanone, 2-hexenal, and pentanal was higher relatively in the *L. plantarum*-12 group (circled region G). These results demonstrated that some aldehydes were formed after fermentation with the addition of *Lactoplantibacillus plantarum*, especially 3-methyl butanal, 2-methyl butanal, 2-hexenal, and pentanal. In the *L. fermentum*-6 group, the peak volume of ethyl octanoate, ethyl hexanoate, heptanal, 3-methyl butanal, 2-pentanone, 3-methyl butanol, 1-hexanol, and 2-heptanone was relatively higher (circled region C); the peak volume of 1-hexanol, E-2-nonenal, 1-heptanol, n-nonanal, E-2-octenal, 2-pentylfuran, 2-heptenal, acetoin, 3-methyl butanal, 2-methyl butanal, 2-heptanone, 2-hexenal, pentanal, 2-pentanone, and ethanol was relatively higher after 6 h of fermentation with *Limosilactobacillus fermentum* (circled region H). These results also showed that some aldehydes were formed after fermentation with the addition of *Lactoplantibacillus fermentum*, especially E-2-nonenal, n-nonanal, E-2-octenal, 3-methyl butanal, and 2-hexenal, and the type of the volatile compounds were the most among *L. plantarum*, *L. fermentum* and *L. actis*. In the *L. lactis*-6 group, the peak volume of acetoin, 2-nonenal, 1-hexanol, 1-hexanol, 2-octenal, 2-heptanone, and 2-heptanone was relatively higher (circled region D); the peak volume of heptanal, 6-methyl-5-hepten-2-one, 1-heptanol, 2-pentylfuran, 2-heptenal, 3-methyl butanal, 2-methyl butanal, 2-heptanone, 2-hexenal, and ethanol was relatively higher after 6 h of fermentation with *Lactococcus lactis* (circled region I). These results meant that some aldehydes and furans were formed after fermentation with the addition of *Lactococcus lactis*, especially 2-pentylfuran, 2-heptenal, 3-methyl butanal, 2-methyl butanal, and heptanal. Combining the above results, the aldehydes increased after fermentation, which is mainly caused by the enzymatic hydrolysis of *Lactobacillus*, or the production of free fatty acids [29]. These aldehydes of low carbon atoms would be conducive to delightful flavor, such as the flavor of fruit, fat or some nuts [30,31]. In the *S. cerevisiae*-6 group, the peak volume of hexyl acetate, 1-heptanol, 2-pentanone, 3-methylbutanol, acetoin, 2-nonenal, 1-hexanol, 2-octenal, 1-pentanol, hexanal, pentanal, ethanol, 2-hexenal, and 2-heptanone was relatively higher (circled region E); the peak volume of ethyl octanoate, 2-methyl propyl acetate, ethyl hexanoate, heptanal, 6-methyl-5-hepten-2-one, 1-hexanol, E-2-nonenal, 1-heptanol, n-nonanal, E-2-octenal, 2-pentylfuran, 2-heptenal, acetoin, 3-methyl butanal, 2-methyl butanal, 2-heptanone, 2-hexenal, pentanal, 2-pentanone, and ethanol was relatively higher after 6 hours of fermentation with *Saccharomyces cerevisiae* (circled region J). These results meant that the types and the peak volume were increased after fermentation with the addition of *Saccharomyces cerevisiae*. While there were fewer types of volatile compounds in the NF group than the groups with added strains, this indicated that the addition of different strains could deeply affect the formation of volatile compounds.

In order to clarify how much each volatile substance contributed to the flavor, ROAV was calculated and values of greater than 0.1 are shown in Table 2. In total, the ROAV of the volatile components (>0.1) were moderately different among the *Lactoplantibacillus plantarum*, *Limosilactobacillus fermentum*, and *Lactococcus lactis* groups at the same stage, and the main volatile compounds were significantly different at different stages. In addition, the main volatile compounds showed higher ROAV in the *S. cerevisiae* group.

In detail, E-2-nonenal dimers (ranging from 23.26% to 35.59%), E-2-nonenal monomers (100%), and 2-methyl butanals (ranging from 13.12% to 17.27%) were the top three main volatile compounds in the five groups at the first stage; 2-methyl butanal (100%), heptanal dimers (ranging from 30.64% to 50.12%), and heptanal monomers (ranging from 30.64% to 50.12%) were the top three main volatile compounds in the five groups at the second stage; E-2-nonenal dimers (ranging from 29.88% to 59.16%), E-2-nonenal monomers (100%), and heptanal dimers (range from 15.13% to 31.37%). As aldehydes had low threshold values, they processed the lower ROAV and were essential compounds in forming the aroma of FFRNs [32]. There were 13 aldehydes in all of the obtained volatile compounds, with 2-methyl butanal, E-2-nonenal, heptanal, and n-nonanal being the main volatile compounds in the entire fermentation process. These aldehydes could give off grassy lily, citrus or fruit smells at their low concentrations [30]. Similarly, there were nine esters among the key volatile compounds, including ethyl hexanoate, ethyl acetate and hexyl acetate, which mainly originated from acids and exude a fruity or fat flavor [31]. 1-Heptanol and 1-hexanol were the only two alcohols in the five groups, while the peak volume alcohols of added mixture strains accounted for about 24.31% in our previous study [6]. The literature had reported that *Saccharomyces cerevisiae* could promote the reactions of acid and active acetaldehyde under acetyl-CoA, which was why the *L. plantarum*, *L. fermentum* and *L. lactis* groups were low in alcohol. Furthermore, 6-methyl-5-hepten-2-one, 2-heptanone dimers, and acetoin were the only two ketones in the five groups, while the peak volume ketones of added mixture strains accounted for about 19.17% in our previous study [6]. Therefore, mixture strains were more beneficial to the degradation of amino acids [32,33].

## 4. Conclusions

The microbial succession and volatile change of FFRNs fermented with four selected strains at 0 h, 6 h, and 12 h stages were studied, and their texture and cooking qualities analyzed. Results showed that the addition of *Lactoplantibacillus plantarum*, *Limosilactobacillus fermentum*, and *Lactococcus lactis* could shorten fermentation time while the addition of *S. cerevisiae* had little effect on fermentation time. Meanwhile, *Lactoplantibacillus plantarum*, *Limosilactobacillus fermentum*, *Lactococcus lactis*, and *Saccharomyces cerevisiae* could proliferate rapidly during the first fermentation stage while their separate additions did not ensure the safety of FFRNs. There were 29 volatile compounds that possessed a ROAV > 0.1, which made a great difference in the flavor formation of FFRN. Notably, there are more volatile components in the S. cerevisiae group than in the *L. plantarum*, *L. fermentum*, and *L. lactis* groups. Finally, the cooking qualities of FFRNs decreased when selected strains were added. Therefore, the addition of a single strain could be taken to improve the cooking qualities and the texture quality of fresh fermented rice noodles. The basic results of this study will provide theoretical support for the research and production of FFRNs in the future.

## Figures and Tables

**Figure 1 foods-12-00961-f001:**
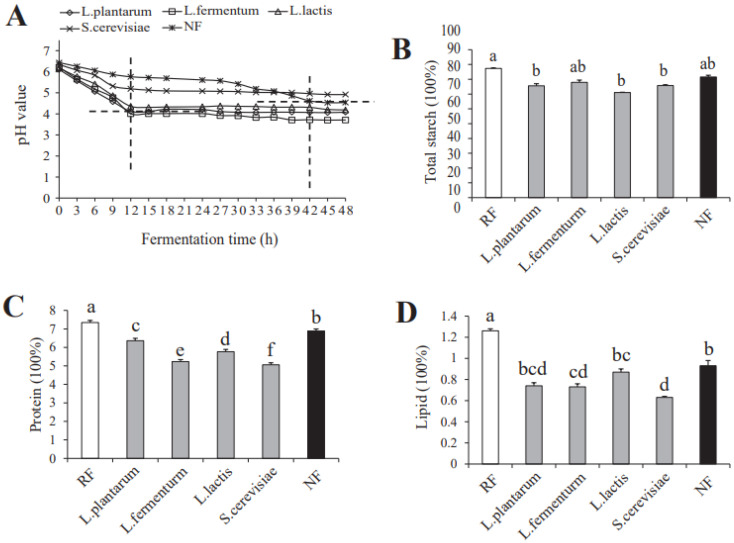
The pH value (**A**) of the fresh fermented rice noodles and the total starch content (**B**), protein (**C**), lipid (**D**) of the RF, *L. plantarum*, *L. fermentum*, *L. lactis*, *S. cerevisiae* and NF rice noodles. Mean values ± standard deviation, *n* = 3. Different superscript letters indicate significant differences at *p* < 0.05 (Tukey Kramer’s multiple comparison post hoc tests). RF means rice flour; *L. plantarum* represents adding *Lactobacillus plantarum*; *L. fermentum* represents adding *Lactobacillus fermentum*; *L. lactis* represents adding *Lactococcus lactis*; *S. cerevisiae* represents adding *Saccharomyces cerevisiae* and NF represents fermented naturally.

**Figure 2 foods-12-00961-f002:**
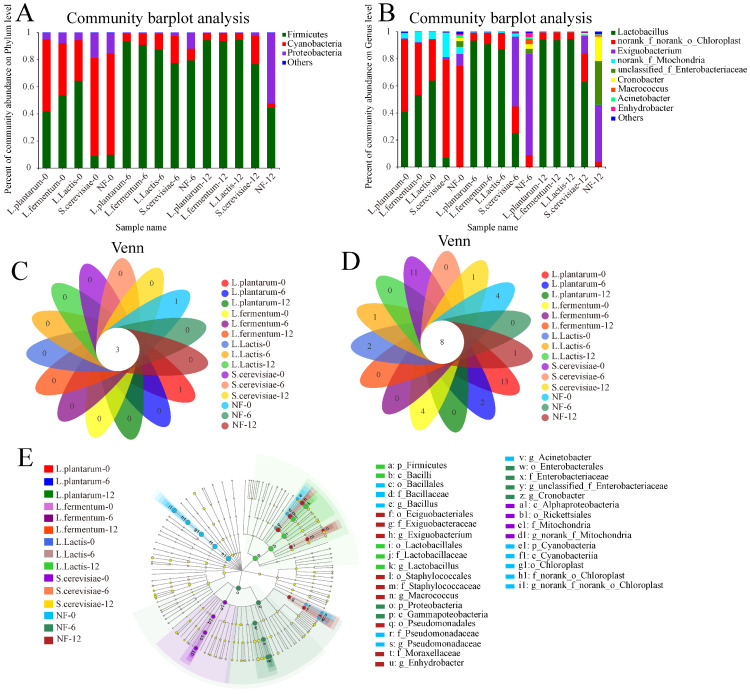
Changes of fresh fermented rice noodles bacterial diversity. (**A**) Relative abundance of bacterial community at the phyla level; (**B**) Relative abundance of bacterial community at the genus level; (**C**) The Veen diagram of bacterial composition the phyla level; (**D**) The Veen diagram of bacterial composition the genus level; (**E**) Linear discriminant analysis effect size (LEfSe)analysis of the bacterial composition. *L. plantarum* represents adding *Lactobacillus plantarum*; *L. fermentum* represents adding *Lactobacillus fermentum*; *L. lactis* represents adding *Lactococcus lactis*; *S. cerevisiae* represents adding *Saccharomyces cerevisiae* and NF represents fermented naturally.

**Figure 3 foods-12-00961-f003:**
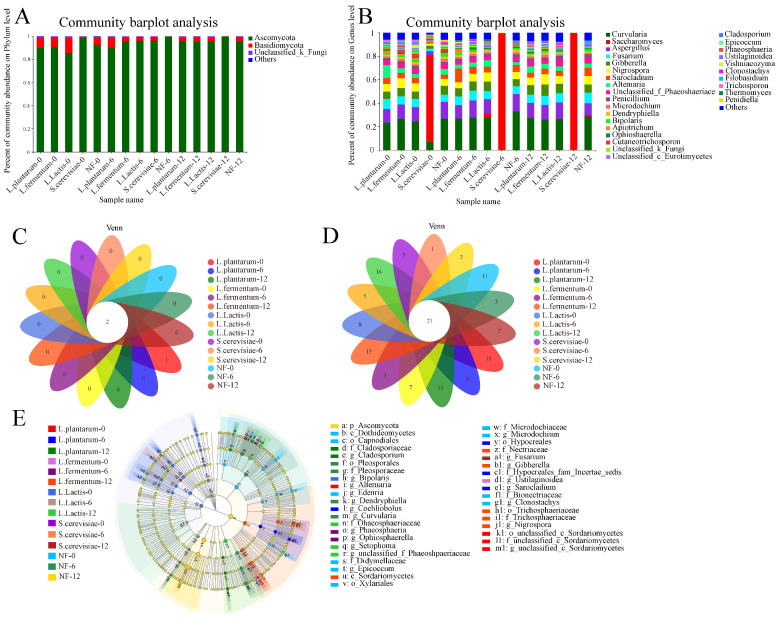
Changes of fresh fermented rice noodles fungal diversity. (**A**) Relative abundance of fungal community at the phyla level; (**B**) Relative abundance of fungal community at the genus level. (**C**) The Veen diagram of fungal composition the phyla level; (**D**) The Veen diagram of fungal composition the genus level; (**E**) Linear discriminant analysis effect size (LEfSe) analysis of the fungal composition. *L. plantarum* represents adding *Lactobacillus plantarum*; *L. fermentum* represents adding *Lactobacillus fermentum*; *L. lactis* represents adding *Lactococcus lactis*; *S. cerevisiae* represents adding *Saccharomyces cerevisiae* and NF represents fermented naturally.

**Figure 4 foods-12-00961-f004:**
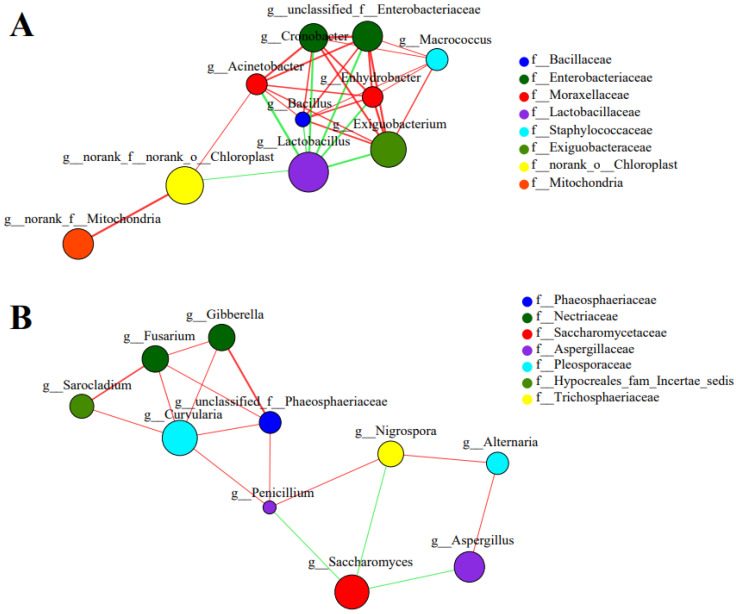
Correlation network of bacterial (**A**) and fungal (**B**) occurrence patterns during fermentation. Only correlations with top 10 were shown. Nodes represent genera and lines represent co-occurrence (R > 0.6, *p* < 0.05, red lines) or co-exclusion (R < −0.6, *p* < 0.05, green lines).

**Figure 5 foods-12-00961-f005:**
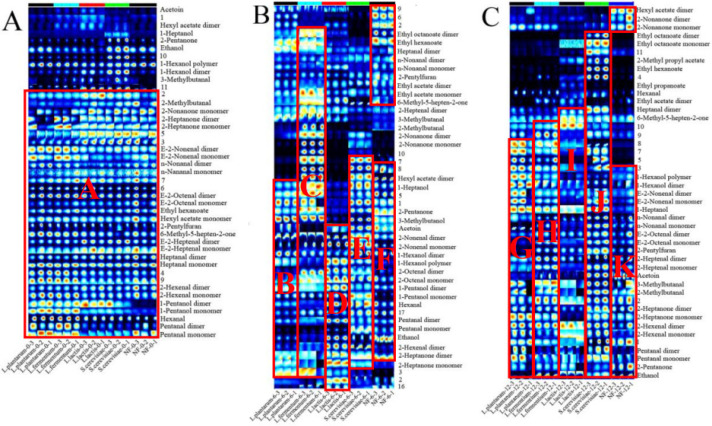
The qualitative analysis topographic plots (**A**–**C**) of fresh fermented rice noodles by HS-GC-IMS. *L. plantarum* represents adding *Lactobacillus plantarum*; *L. fermentum* represents adding *Lactobacillus fermentum*; *L. lactis* represents adding *Lactococcus lactis*; *S. cerevisiae* represents adding *Saccharomyces cerevisiae* and NF represents fermented naturally.

**Table 1 foods-12-00961-t001:** Cooking properties and texture profile analysis (TPA) of five fresh fermented rice noodles.

	*L. plantarum*	*L. fermentum*	*L. lactis*	*S. cerevisiae*	NF
Cooking Properties					
Water absorption rate (%)	14.25 ± 1.03 ^cd^	15.22 ± 1.05 ^c^	13.37 ± 1.35 ^d^	19.16 ± 0.89 ^a^	16.63 ± 1.32 ^b^
Cooking loss (%)	2.73 ± 0.33 ^c^	2.92 ± 0.19 ^b^	2.66 ± 0.13 ^d^	3.05 ± 0.16 ^ab^	3.11 ± 0.11 ^a^
Turbidity (at 675 nm)	0.29 ± 0.07 ^b^	0.31 ± 0.02 ^b^	0.28 ± 0.05 ^a^	0.34 ± 0.02 ^b^	0.36 ± 0.04 ^b^
TPA					
Hardness (g)	1827 ± 219 ^b^	1616 ± 147 ^c^	1980 ± 207 ^a^	1555 ± 232 ^c^	1186 ± 178 ^d^
Adhesiveness (g. s)	−28.46 ± 4.16 ^cd^	−29.53 ± 3.36 ^bc^	−28.84 ± 3.24 ^d^	−39.13 ± 4.67 ^ab^	−28.08 ± 2.82 ^a^
Resilience (%)	50.86 ± 2.32 ^c^	52.28 ± 2.29 ^b^	49.16 ± 3.59 ^c^	57.31 ± 3.69 ^a^	40.21 ± 2.43 ^d^
Springiness (%)	94.23 ± 1.18 ^ab^	94.52 ± 1.66 ^ab^	94.61 ± 1.62 ^ab^	93.29 ± 0.64 ^c^	95.67 ± 2.13 ^a^
Cohesion	0.84 ± 0.01 ^ab^	0.83 ± 0.01 ^bc^	0.86 ± 0.01 ^a^	0.83 ± 0.01 ^bc^	0.84 ± 0.02 ^ab^
Chewiness	1421 ± 129 ^b^	1244 ± 113 ^c^	1619 ± 134 ^a^	1017 ± 121 ^d^	1036 ± 136 ^d^

*L. plantarum* represents adding *Lactobacillus plantarum*; *L. fermentum* represents adding *Lactobacillus fermentum*; *L. lactis* represents adding *Lactococcus lactis*; *S. cerevisiae* represents adding *Saccharomyces cerevisiae*. Means values ± standard deviation, *n* = 3. Different superscript letters indicate significant differences at *p* < 0.05 (Tukey Kramer’s multiple comparison post hoc tests).

**Table 2 foods-12-00961-t002:** Relative odor activity value of volatile compounds detected by HS-GC-IMS in fresh fermented rice noodles.

		Relative Odor Activity Value (ROAV)															
		0 h						6 h					Rice Noodle				
Compounds	Describe	*L. plantarum*		*L. fermentum*	*L. lactis*	*S. cerevisiae*	NF	*L. plantarum*	*L. fermentum*	*L. lactis*	*S. cerevisiae*	NF	*L. plantarum*	*L. fermentum*	*L. lactis*	*S. cerevisiae*	NF
E-2-Nonenal dimer	Orris-like ^a^	0.0009 ^b^	26.93	29.12	23.26	35.59	21.82	-	-	-	-	-	47.21	42.66	42.49	59.16	29.88
E-2-Nonenal monomer	Orris-like ^a^	0.0009 ^b^	100	100	100	100	100	-	-	-	-	-	100	100	100	100	100
2-Methyl butanal	Chocolate-like ^a^	0.001 ^b^	16.96	14.23	13.58	13.27	13.12	100	100	100	100	100	16.01	12.53	19.72	16.33	16.24
Heptanal dimer	Fatty ^a^	0.0028 ^b^	6.01	6.53	6.72	6.33	6.24	41.38	35.04	30.64	41.43	50.12	22.37	29.01	31.37	15.13	23.70
Heptanal monomer	Fatty ^a^	0.0028 ^b^	1.35	1.31	1.98	2.19	0.96	22.27	29.00	21.37	35.13	23.70	-	-	-	-	-
n-Nonanal dimer	Orange, rose ^a^	0.0031 ^b^	7.24	9.31	6.77	7.96	5.39	13.11	12.13	16.35	17.26	16.61	-	-	-	-	-
n-Nonanal monomer	Orange, rose ^a^	0.0031 ^b^	3.21	2.52	3.01	2.44	2.36	1.87	1.67	1.04	1.39	0.80	-	-	-	-	-
3-Methyl butanal	Almond ^a^	0.029 ^b^	0.21	0.37	0.41	0.41	<0.1	0.59	1.46	0.83	0.39	0.11	0.19	0.15	0.21	0.14	0.13
2-Hexenal monomer	Sweet ^a^	0.03 ^b^	0.64	0.52	0.59	0.78	0.14	-	-	-	-	-	5.9	5.41	6.77	6.63	4.61
2-Hexenal dimer	Sweet ^a^	0.03 ^b^	0.11	<0.10	0.11	0.12	0.11	0.1	0.11	0.22	0.24	0.11	0.12	0.12	0.14	0.13	0.09
Hexanal	Grassy, fruity ^a^	0.0615 ^b^	0.62	0.52	0.33	0.93	0.65	0.47	0.39	0.32	0.70	0.09	2.80	5.45	3.21	7.21	0.19
E-2-Octenal monomer	Green-leafy ^a^	0.25 ^b^	<0.10	0.10	<0.10	<0.10	<0.10	-	-	-	-	-	<0.10	<0.10	<0.10	<0.10	<0.10
E-2-Octenal dimer	Green-leafy ^a^	0.25 ^b^	<0.10	<0.10	0.11	0.13	<0.10	-	-	-	-	-	<0.10	<0.10	<0.10	<0.10	<0.10
Ethyl acetate dimer	Pineapple-like ^a^	0.005 ^b^	2.04	2.33	2.32	5.23	2.95	8.91	8.13	9.29	11.45	8.74	11.14	15.27	14.52	13.10	5.37
Ethyl acetate monomer	Pineapple-like ^a^	0.005 ^b^	-	-	-	-	-	2.50	2.74	2.71	3.02	5.13	-	-	-	-	-
Ethyl hexanoate	Banana-like ^a^	0.005 ^b^	4.95	4.23	3.40	3.53	3.12	4.65	3.81	4.94	4.84	4.79	-	-	-	-	-
Ethyl octanoate dimer	Floral ^a^	0.0193 ^b^	-	-	-	-	-	0.13	0.12	0.33	2.98	0.15	-	-	-	-	-
Ethyl propanoate	Rum a	0.01 ^b^	-	-	-	-	-	-	-	-	-	-	9.64	10.29	12.91	13.93	2.91
Ethyl octanoate monomer	Floral ^a^	0.0193 ^b^	-	-	-	-	-	-	-	-	-	-	6.12	9.12	7.17		
Ethyl octanoate dimer	Floral ^a^	0.0193 ^b^	-	-	-	-	-	-	-	-	-	-	5.36	4.50	1.37		
Hexyl acetate monomer	Cherry, pear, floral ^a^	0.115 ^b^	0.11	0.17	0.37	1.90	0.12	0.10	<0.10	<0.10	<0.10	<0.10	0	0	0		
Hexyl acetate dimer	Cherry, pear, floral ^a^	0.115 ^b^	<0.10	<0.10	<0.10	<0.10	<0.10	<0.10	<0.10	<0.10	<0.10	<0.10	0.13	0.18	0.10		
6-Methyl-5-hepten-2-one	Citrus-like ^a^	0.068 ^b^	0.12	0.10	0.13	0.14	0.12	0.1	<0.1	0.16	0.17	0.10	0.12	0.11	0.14		
2-Heptanone dimer	Fruity ^a^	0.14 ^b^	<0.10	<0.10	0.11	0.13	<0.10	<0.10	<0.10	<0.10	<0.1	<0.1	<0.10	<0.10	0.11		
2-Heptanone monomer	Fruity ^a^	0.14 ^b^	<0.10	<0.10	<0.10	0.33	<0.1	0.29	0.45	0.31	0.56	0.17	0.19	0.36	19.78		
Acetoin	Woody ^a^	0.014 ^b^	1.18	1.18	1.17	1.77	1.12	1.48	1.52	0.38	2.24	0.56	3.24	5.76	4.43		
1-Heptanol	Fragrant ^a^	0.023 ^b^	<0.10	<0.10	<0.10	<0.10	0.10	<0.10	<0.10	0.15	1.27	0.55	0.36	0.97	0.12		
1-Hexanol dimer	Aromatic ^a^	0.04 ^b^	0.79	0.73	0.60	0.69	0.62	0.26	0.17	0.30	0.36	<0.10	0.18	0.21	0.26		
2-Pentylfuran	Vegetable ^a^	0.0058 ^b^	1.42	1.37	2.01	1.67	1.25	1.07	1.10	2.12	2.09	1.62	3.00	2.79	2.90		

*L. plantarum* represents adding *Lactobacillus plantarum*; *L. fermentum* represents adding *Lactobacillus fermentum*; *L. lactis* represents adding *Lactococcus lactis*; *S. cerevisiae* represents adding *Saccharomyces cerevisiae*. ^a^ represents odor threshold values according to Van Gemert (2015). ^b^ represents odor description according to Burdock, G.A. (2010).

## Data Availability

Data is contained within the article.

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
