# Peer review of "Further Interpretation of the Volatile, Microbial Community and Edible Quality of Fresh Fermented Rice Noodles with Different Selected Strains"

_foods, 2023, doi:10.3390/foods12050961_

Round 1

Reviewer 1 Report

Dear Authors

Present manuscript "Further interpretation of the volatile, microbial community and edible quality of fresh fermented rice noodles with different selected strains" is well -structured but I found some similarty between this paper and "Aixia Wang, Tianzhen Xiao, Huihan Xi, Wanyu Qin et al. "Edible qualities, microbial compositions and volatile compounds in fresh fermented rice noodles fermented with different starter cultures", Food Research International, 2022".

So please

1.Clearify the main differences between them. infact you'd better state the innovation of your work in Introduction section.

2. Abstract should be supported by numerical report of the results

3. Reduce similarity of text

Author Response

Dear reviewer,

Thanks for your suggestion. We have explained, point by point, the details
of the revisions to the manuscript and the responses to the referees’
comments. The responses file have attached.

Reviewer 2 Report

foods-2180110

Further interpretation of the volatile, microbial community and edible quality of fresh fermented rice noodles with different selected strains

Aixia Wang , Songfeng Xie , Zengrun Xia , Fengzhong Wang , Li-Tao Tong  

Notes for authors:

·         Please describe the purpose of the research in more detail in the introduction. Influence of taste profile on consumer preferences and relationship with different strains of bacteria.

·         Line 81, 86  Give more details to describe the methods you use.

·         Indicate the methods (line 94)of these determinations and briefly describe them (line 95, 98, 122). ·         Describe the preparation of the sample for the analysis of volatile compounds. ·         The statistical methods used and the method of presenting the results should be described in detail. ·         During the fermentation process, many changes occur in the chemical compounds contained in the material. in order to support the thesis of a change in reaction associated with the growth of lactic acid bacteria, it would be useful to determine the content of this acid or titratable acidity. ·         Line 159 Was the water content of the fermented material measured? ·         Please discuss the obtained results with other items available in the literature. ·         lines 39-42 not completed                  The authors described the important topic of rice noodle fermentation using different strains of bacteria: Limosilactobacillus fermentum, Lactoplantibacillus plantarum, Lactococcus lactis and Saccharomyces cerevisiae. Manuscrypt requires the addition of a description of the analytical methods used and discussion with other authors dealing with this topic. Language proofreading is also required.

Author Response

(The authors gave the same response as above.)

Reviewer 3 Report

Overall, the article is well presented and discusses interesting results in the production of fresh fermented rice noodle (FFRN). Follows only minor corrections.

Minor corrections:

1) You didn't specify the NF, as you did with RF (rice flour);

2) FOrmat in italic the "L" of "L. fermentum"; the "S" of "S. cerevisiae", and so on (page 4, lines 152-155);

3) Format in italic species name written on Table 1;

4) I suggests increase the font size of Figures 2 and 3;

5) You didn't specify the LEfSe;

6) Exclude the italic formation of word "adding" (page 6, lines 209-210 and page 8, lines: 250-252);

7) Change "acetyl-coA" to "acetyl-CoA" ("C" in uppercase); 

8) This sentence is strange:

"Notably, the volatile components of S. cerevisiae group more than the volatile components of L. plantarum, L. fermentum and L. lactis groups" (page 14, lines 34-35, in Conclusisons)

I suggests:

"Notably, there is more volatile components of S. cerevisiae group than volatile components of L. plantarum, L. fermentum and L. lactis groups"

Author Response

(The authors gave the same response as above.)

Reviewer 4 Report

This paper is unique contribution to better understanding fermented rice noodle production and quality.

Detail comments with needed corrections are noted in pdf file.

Author Response

(The authors gave the same response as above.)

Round 2

Reviewer 1 Report

All questions were answered. Now it is suitable for publication

Reviewer 4 Report

Quality of the reseach is improved by implemented corrections